# DNA Barcoding of Chironomid Larvae (Diptera: Chironomidae) from Large Rivers in South Korea to Facilitate Freshwater Biomonitoring and Public Health Surveillance

**DOI:** 10.3390/ijerph191912035

**Published:** 2022-09-23

**Authors:** Hyo Jeong Kang, Min Jeong Baek, Ji Hyoun Kang, Yeon Jae Bae

**Affiliations:** 1Department of Life Science, Graduate School, Korea University, Seoul 02841, Korea; 2National Institute of Biological Resources, Incheon 22689, Korea; 3Korean Entomological Institute, Korea University, Seoul 02841, Korea; 4Department of Environmental Science and Ecological Engineering, College of Life Sciences, Korea University, Seoul 02841, Korea

**Keywords:** large river Chironomidae, DNA barcode library, larval-adult association, rapid identification protocol, South Korean rivers

## Abstract

Chironomid larvae are among the dominant benthic macroinvertebrates in all types of water systems in South Korea. They may pass through pipes in rivers (raw water) and occur in drinking water, thus creating public health issues. However, little is known about the larval stages of chironomids in large South Korean rivers. Therefore, we examined larval–adult associations in chironomids inhabiting major rivers used as water sources. The larvae were collected in 2015 and 2016 from nine locations along the four largest rivers in South Korea using a Ponar grab. Cytochrome oxidase subunit I (*COI*) sequences were generated from the larval specimens, and the species were identified by comparing these sequences to those in a newly constructed DNA barcode library of Chironomidae in South Korea. The samples from the four rivers yielded 61 mitochondrial *COI* sequences belonging to 18 species, including *Hydrobaenus kondoi* Saether, 1989, which was reported for the first time in the Korean Peninsula. Further, morphological identification of the larvae was conducted, and a pictorial taxonomic key to Chironomidae species in large rivers in South Korea was developed to facilitate freshwater biomonitoring research. Finally, an action flow chart was created for the rapid identification of chironomid larvae in infested drinking water or water purification facilities.

## 1. Introduction

Rivers are an important source of freshwater for humans. In South Korea, the main sources of industrial and domestic water supply are the country’s four largest rivers: the Han (HR), Geum (GR), Yeongsan (YR), and Nakdong Rivers (NR) [1]. However, as the country develops, anthropogenic activities (e.g., dam construction) have negatively impacted South Korea’s river ecosystems through increased channelization, eutrophication, and the increase in lentic areas [2]. These activities, in combination with rising water temperatures due to climate change, have promoted the mass emergence of sporadic/nuisance insect species inhabiting rivers and streams. Among these, chironomids (Diptera: Chironomidae) have become one of the dominant invertebrate groups in South Korean large rivers, and along with oligochaete worms, they account for over 90% of the benthic macroinvertebrate fauna in terms of richness and abundance, particularly in the lentic areas of large river systems [3,4,5]. Most cases of sporadic/nuisance insect emergence in South Korean rivers can be attributed to riverine chironomids.

Owing to the massive occurrence and abundance of riverine chironomids in South Korea, the invasion of tap water and drinking water purification facilities by their larvae has recently become a key social issue [6]; numerous cases of larval infestation of tap water have been reported in many large and small cities, including Incheon, Seoul, Busan, and Jeju [6,7], raising national concerns regarding water safety. However, in severe cases of larval infestation in tap water and water purification facilities, most local governments have not been able to appropriately respond due to lack of information on chironomid larvae.

Chironomid larvae are widely used in freshwater biomonitoring due to their high abundance, rich species diversity, high pollution tolerance, overlapping life cycles, low larval mobility, and ease of sampling [8,9]. Moreover, they are valuable models for toxicity tests under specific environmental conditions, such as assays of toxic sediment stress and microplastic concentrations in freshwater [10,11,12,13]; however, despite their rich species diversity, chironomids are often neglected in aquatic biodiversity and biomonitoring studies due to the lack of knowledge on larval identification.

As is the case with other dipteran groups, many chironomid species are difficult to discriminate at the larval stage, and chironomid taxonomy is based primarily on adult male morphology, such as the characteristics of the male genital hypopygium [14]. Furthermore, conventional approaches to establish adult–larval associations in Diptera typically involve rearing larvae or collecting pupal exuviae to associate with adults, but these methods also have significant technical limitations: larval rearing can be inefficient in certain groups, and generally requires specific conditions (e.g., temperature, food, and photoperiod), whereas pupal sampling is limited by short emergence periods [15,16,17].

However, the identification of chironomid larvae is critical for rapidly responding to public health issues such as larval infestation in drinking water as well as for supporting freshwater biomonitoring research. In this context, recently developed DNA barcodes for adult chironomids using mitochondrial *COI* genes [18] have rendered adult–larval association practicable. Therefore, in this study, we aimed (i) to correctly identify chironomid larvae inhabiting large rivers in South Korea using morphology and DNA barcodes, (ii) to create a pictorial key to facilitate the identification of chironomid larvae by researchers and the public, and (iii) to develop a protocol for public agencies and stakeholders, which will allow the rapid and accurate examination of invertebrate species found in drinking water.

## 2. Materials and Methods

### 2.1. Larval Sampling and Morphological Identification

In 2015 and 2016, chironomid larvae were collected from nine locations distributed across South Korea’s four largest rivers (HR, three sites; GR, three sites; NR, two sites; and YR, one site; Figure 1). The larvae were collected from lentic areas (2–10 m deep; 20 samples in total per site) approximately 0.3–1 km upstream of weirs using a Ponar grab (15 × 15 cm) on a boat. River substrates were relatively homogenous mixtures of sand (20–40%) and silt (60–80%) at all sites, except those along the HR, where the proportion of sand was higher (50–70%). Each sample was passed through a 0.5 mm sieve, and the recovered larvae were preserved in 80% ethanol. The preserved larval specimens were separated into morpho-species, photographed (Zeiss Stereo Discovery V12 microscope; Carl Zeiss Microscope GmbH, Jena, Germany) to examine thoracic and abdominal setae and tubules, and then identified to the genus or subfamily level using available identification keys and references [19,20,21,22]. For each specimen, the head and abdominal segments VII–IX were carefully dissected under a stereomicroscope (SZX7; Olympus, Tokyo, Japan) and placed in Hoyer’s medium for slide preparation [23]. Additionally, the larvae’s mouthparts were drawn using a drawing tube attached to an optical microscope (BX53; Olympus, Tokyo, Japan), and larval images were digitized using Adobe Illustra-tor CS6 (Adobe Systems Inc., San Jose, CA, USA). The remaining body parts were preserved in absolute ethanol at −20 °C for subsequent DNA extraction.

### 2.2. Molecular Analysis

To reduce the risk of contamination, the specimens were placed in Petri dishes containing 1 mL of 80% ethanol. We dissected, from the thorax I–III to the abdomen, using a needle to reveal the gut, which was carefully removed from the specimen using forceps under a stereomicroscope (SZX7; Olympus, Tokyo, Japan). Total DNA was extracted from thorax segments I–III or abdominal tissue using the DNeasy Blood and Tissue Kit (Qiagen, Hilden, Germany) following the manufacturer’s instructions. Moreover, a 658-bp fragment of the mitochondrial cytochrome oxidase I (*COI*) gene was amplified by polymerase chain reaction (PCR) using the AccuPower PCR premix (Bioneer, Daejeon, South Korea) and previously published universal primers (LCO1490 and HCO2198) [24]. Amplification was performed in 20 μL reactions under the following conditions: 94 °C for 5 min, 35 cycles at 94 °C for 30 s, 48 °C for 1 min, and 72 °C for 1 min 30 s, and 72 °C for 10 min [25]. The resulting PCR products were purified and sequenced according to the methodology described by Kang et al. [18], and the sequences were deposited in GenBank (accession numbers OP381663–OP381723).

To identify the chironomids at the species level, the *COI* sequences obtained from larval specimens were compared against the adult reference sequences from NCBI and our previous studies [18], and the sequences were then aligned using the CLC Main Workbench (version 7.8.1; CLC bio, Aarhus, Denmark). The results were crosschecked using the ClustalW algorithm in MEGA 7.0., and a neighbor-joining (NJ) analysis was conducted using MEGA 7.0 to examine the relationships among specimens [26,27].

## 3. Results

### 3.1. Specimen Identification

A total of 61 *COI* sequences were successfully generated from the larval specimens (Figure 2), and in the NJ analysis, most of the gene’s conspecific sequences were grouped with 100% bootstrap support. Of the 61 sequences, 56 were assigned to 14 species, and one (H20L) was identified at a presumed genus level (*Stenochironomus* Kieffer, 1919). The remaining four sequences (H01L, N28L, N29L, and N31L) were assigned to three undescribed species. Of the 14 identified species, 11 belonged to nine genera in the Chironominae subfamily, two belonged to two genera in the Orthocladiinae subfamily, and one belonged to a genus in the Tanypodinae subfamily (Table 1). Furthermore, the *COI* sequences of specimens identified as *Benthalia carbonaria* (Meigen, 1804) formed two clades which differed by a maximum of 4.2% (K2P-distances) or 77 nucleotide sites. These diverging specimens were collected from the GR and NR populations (Figure 2).

### 3.2. Morphological Key and Action Protocol

The larvae were morphologically identified and a pictorial key was created based on their diagnostic characteristics (Figure 3), including head capsule features such as eye spots, antennae, and mouthparts; tubules on abdominal segments VII–VIII; and armature of segment IX. Based on these key characteristics, taxa were identified at subfamily, tribe, genus, and species levels. Furthermore, an action protocol was formulated for public use in order to enable a rapid response to environmental issues, such as larval infestation of drinking water (Figure 4). The mechanism underlying the protocol was as follows: when small insects such as midges are detected in drinking water, public agencies are notified, and stakeholders then collect samples that are sent to identification experts. Specimen collection is important for quickly identifying species according to the provided protocol, and therefore the flow chart of the protocol includes a simple morphological identification process that uses the pictorial key (Figure 3) and a DNA barcode analysis should larval material be damaged or incomplete.

## 4. Discussion

The chironomid fauna of large rivers in South Korea, which are for the source of most of the freshwater supply in the country, remains virtually unknown due to difficulties in the sampling of deep river beds and identification of specimens because of limited knowledge on larval morphology. Therefore, we used a previously developed chironomid DNA barcode library [18] in this study to identify, for the first time, larval specimens of 18 chironomid species inhabiting four major rivers in South Korea, which account for approximately 45% of all known adult chironomid species found in these rivers [18].

Currently, DNA barcoding is a well-established tool in taxonomy and ecology, which has been widely applied for the identification of pests, invasive species, and foreign insect materials in food, in addition to various other applications in the public health sector [28,29,30,31]. Although the present study incorporated sequences from GenBank, we used our own species identification data based on adult morphology, as well as DNA sequences of adult and larval specimens from the same sampling sites, to improve the accuracy of species identification [18]. In this study, larval and adult specimens were correctly differentiated by (1) identifying adult males based on morphological traits, specifically features of their genital structures; (2) constructing phylogenetic trees based on *COI* sequences of adult male specimens to clearly delineate the species category; and (3) assigning larval sequences to species categories, established according to the phylogenetic analysis. Nonetheless, NCBI reference data should be used with caution as sequences may be derived from different geographic regions or misidentified specimens [32,33].

In addition, as shown in the present study, the high-resolution identification of larval specimens can be achieved using a combination of DNA barcoding and morphological analyses. For instance, the approach we adopted allowed us to identify for the first time a chironomid species complex that was previously unidentifiable from larval specimens and lacked a clearly delineated species category, which was *Hydrobaenus kondoi* Saether, 1989. Moreover, we noted morphological deviations from the data recorded in the Holarctic key during genus identification using the general taxonomic key [19]; for instance, a record for a larva belonging to the genus *Dicrotendipes* in the Holarctic key indicated that it possessed a pair of ventral tubes, although two specimens collected from HR in this study did not exhibit this feature. These differences may represent intraspecific or intrageneric variations in larval morphology; therefore, further comprehensive evaluation of larval and adult material, as well as DNA sequencing data, will most likely help define chironomid taxa.

Furthermore, in line with our observations, Kim et al. [34] reported that the genera *Chironomus*, *Dicrotendipes*, *Glyptotendipes*, *Michrochironomus*, *Polypedilum*, *Stictochironomus*, *Tanytarsus*, and *Propsilocerus* were most abundant in large South Korean rivers. Individuals in these genera can thrive in large rivers because of their adaptive larval and pupal traits such as hemoglobin production, and specific behaviors such as the construction of tubes and burrows in fine sediments [35,36,37]. Moreover, most of the species identified in our study are resistant to eutrophication, and some genera are highly resistant to hypoxia, a frequent phenomenon occurring in polluted streams and benthic zones in lentic areas of large rivers in Northeast Asia. Furthermore, our results suggest that larval species inhabiting large South Korean rivers may be detected in tap water and water purification facilities, because water resources are supplied from the impoundment areas of dams and weirs of these rivers.

Infestation of tap water by chironomid larvae was first reported in Incheon, South Korea in July 2020 [6], which escalated to a nationwide hot topic. Since then, local governments throughout the country have devoted close attention to drinking water purification facilities and supply systems, with the inclusion of “larval infestation” as a new criterion in the drinking water quality checklist. Indeed, larval infestation of tap water has been reported recurrently since the 1940s in other countries, such as Germany, Great Britain, the United States, and Israel [38,39,40,41,42].

The new pictorial key and protocol developed for the identification of chironomid species found in large South Korean rivers presented herein (Figure 3 and Figure 4) are expected to facilitate the identification of larvae associated with infestation problems and, along with the attached taxonomic key (Appendix A), they may enable the biomonitoring of large rivers in East Asia [43] as practiced for larval chironomids in Europe, North America, and Australia [19,20,21,22]. Furthermore, the action protocol formulated here facilitates rapid identification of chironomid larvae in infested drinking water (Figure 4).

## 5. Conclusions

Through the comprehensive sampling of chironomid larvae from four large rivers in South Korea and the use of a recently constructed DNA barcode library of Korean Chironomidae, this study identified 18 species within this family and successfully associated larval morphotypes with known adult species for the first time. Moreover, our newly developed pictorial key to chironomid species in large South Korean rivers may facilitate biomonitoring of large rivers in Northeast Asia, and our action protocol for larval identification can be used by the public to enable a rapid response to environmental issues such as larval infestation of drinking water, helping alleviate a problem that has greatly affected countries such as Korea in the past.

## Figures and Tables

**Figure 1 ijerph-19-12035-f001:**
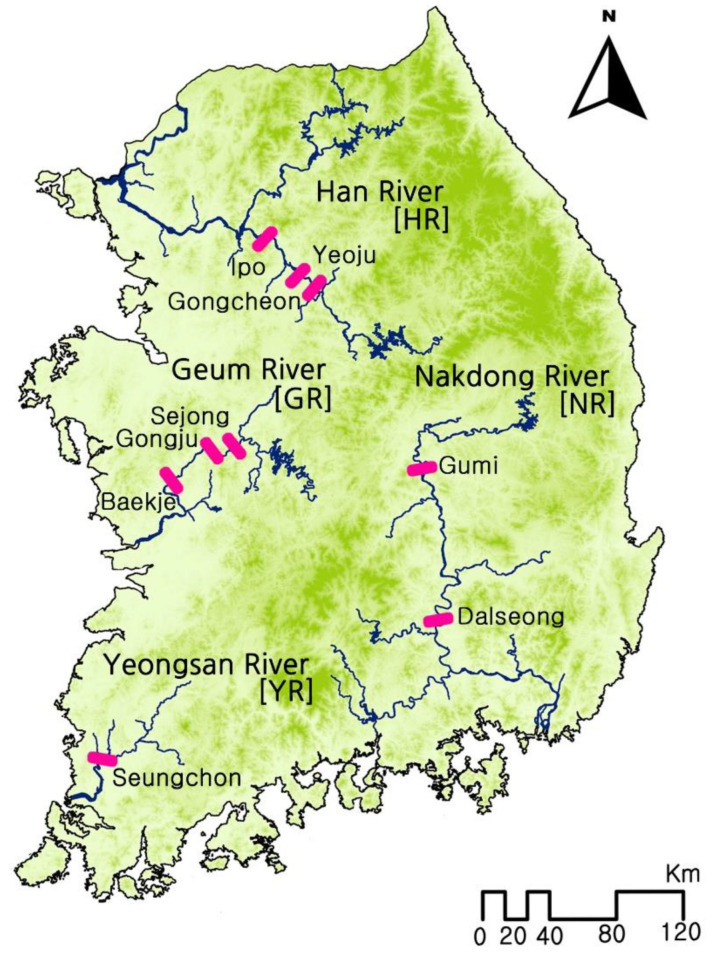
Sampling sites along the four largest rivers in South Korea. Sampling was conducted at 0.3–1 km upstream of weirs (pink marks) at each site.

**Figure 2 ijerph-19-12035-f002:**
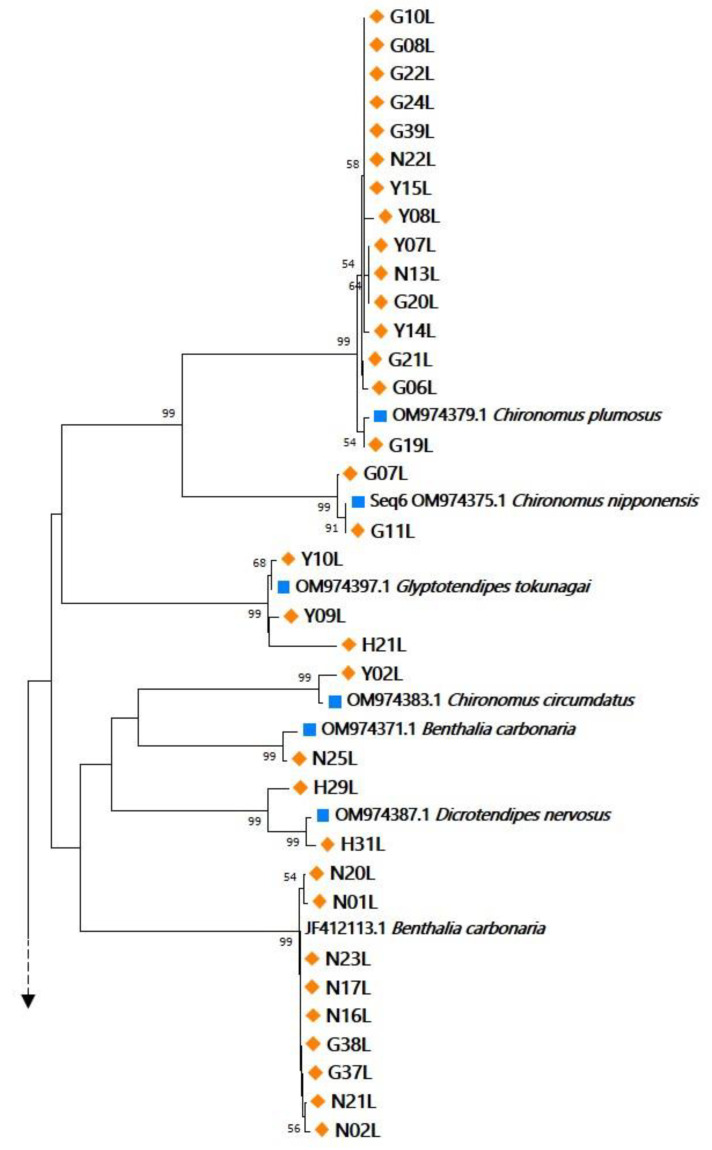
Neighbor-joining tree of 79 cytochrome oxidase I (*COI*) sequences from larval and adult chironomid specimens. Sequences from larval and adult specimens are marked using orange diamonds and blue squares, respectively, and the other of adult reference sequences from NCBI are unmarked.

**Figure 3 ijerph-19-12035-f003:**
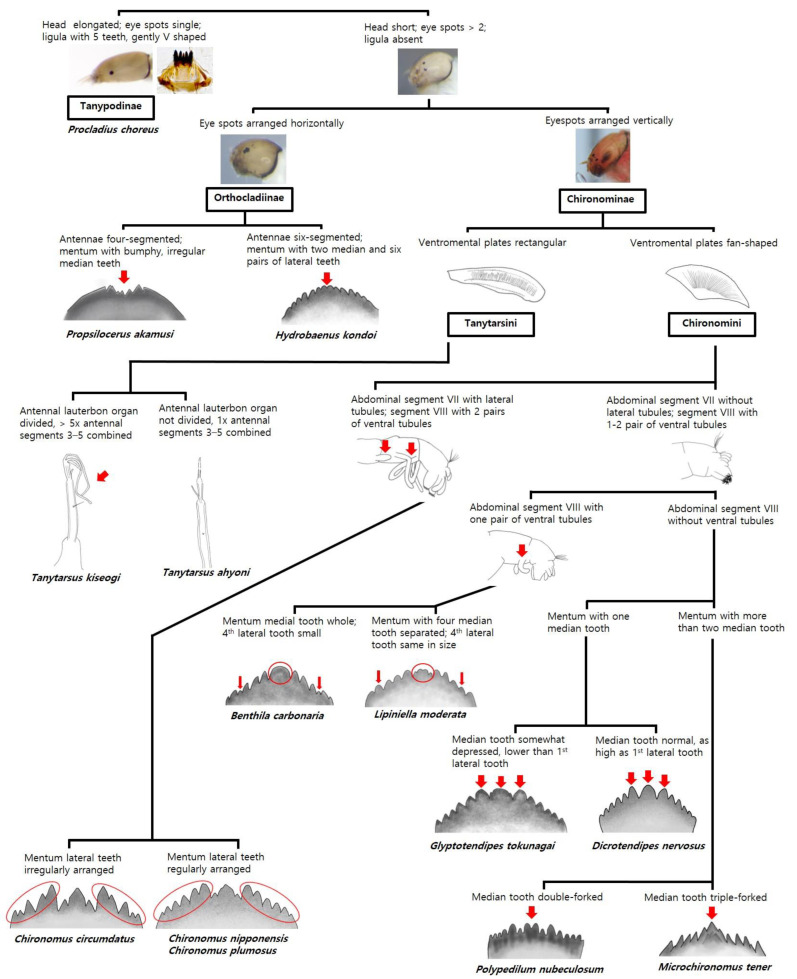
Pictorial key to chironomid larvae inhabiting large rivers in South Korea.

**Figure 4 ijerph-19-12035-f004:**
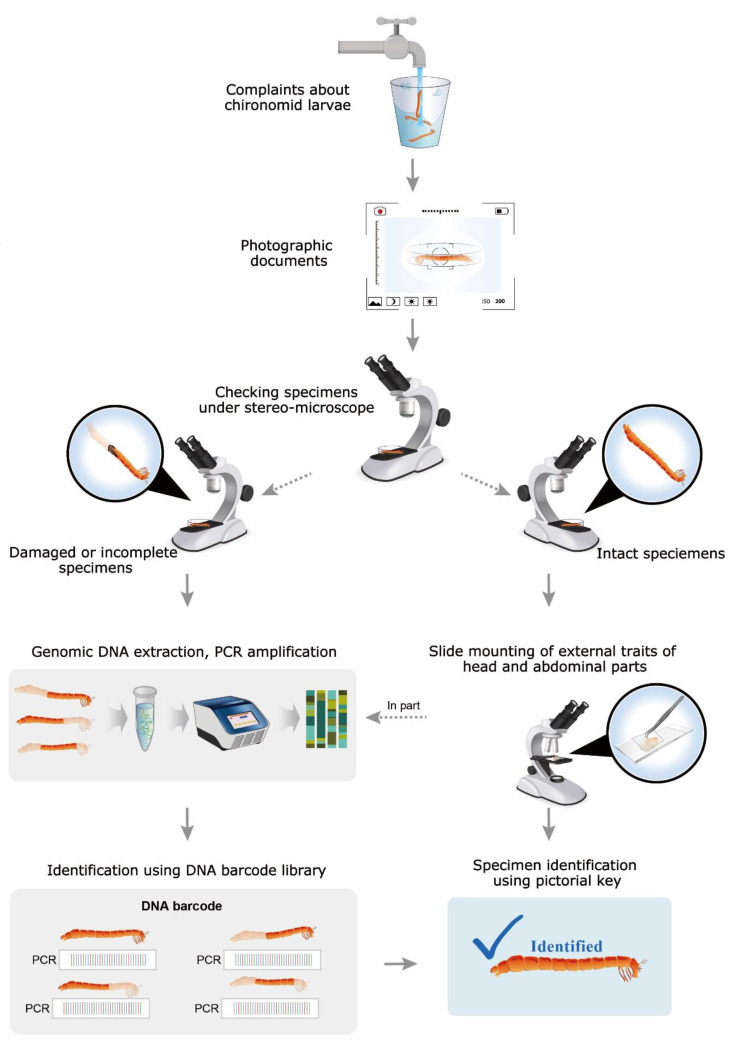
Action protocol for rapid identification of chironomid larvae to respond to environmental issues.

**Table 1 ijerph-19-12035-t001:** List of larval Chironomidae taxa collected from four large rivers in South Korea and identified using a DNA barcode library in this study.

Subfamily	Species	Accession Number
Tanypodinae	*Procladius choreus* (Meigen, 1804)	OM974448 ^†^, OP381670, OP381671, OP381677, OP381678
Orthocladiinae	*Hydrobaenus kondoi* Saether, 1989 *	KP902814 ^†^, OP381696
	*Propsilocerus akamusi* (Tokunaga, 1938)	JN887116 ^†^, OP381672, OP381685, OP381686, OP381690, OP381691, OP381692, OP381698, OP381700, OP381718
Chironominae	*Benthalia carbonaria* (Meigen, 1804)	OM974371 ^†^, JF412113 ^†^, OP381680, OP381681, OP381682, OP381683, OP381684, OP381687, OP381688, OP381693, OP381694, OP381704, OP381706
	*Chironomus circumdatus* (Kieffer, 1916)	OM974383 ^†^, OP381679
	*Chironomus nipponensis* Tokunaga, 1940	OM974375 ^†^, OP381719, OP381722
	*Chironomus plumosus* (Linnaeus, 1758)	OM974379 ^†^, OP381668, OP381669, OP381675, OP381676, OP381689, OP381705, OP381713, OP381714, OP381715, OP381716, OP381717, OP381720, OP381721, OP381723
	*Dicrotendipes nervosus* (Staeger, 1839)	OM974387 ^†^, OP381695, OP381697
	*Glyptotendipes tokunagai* Sasa, 1979	OM974397 ^†^, OP381673, OP381674, OP381703,
	*Microchironomus tener* (Kieffer, 1918)	OM974400 ^†^, OP381711, OP381712
	*Lipiniella moderata* Kalugina, 1970	OM974372 ^†^, OP381707, OP381708, OP381709, OP381710
	*Polypedilum nubeculosum* (Meigen, 1804)	OM974421 ^†^, OP381699
	*Stenochironomus* sp.	OP381666
	*Tanytarsus ahyoni* Ree & Jeong, 2010	KT613731 ^†^, OP381702
	*Tanytarsus kiseogi* Ree & Jeong, 2010	JF412169 ^†^, OP381701
	Chironominae sp. 1	OP381663
	Chironominae sp. 2	OP381664, OP381665
	Chironominae sp. 3	OP381667

Asterisks (*) indicate species recorded as new to the South Korean fauna. GenBank accession numbers with daggers (†) indicate the specimens whose sequences were retrieved from GenBank.

## Data Availability

The data presented in this manuscript are available from the authors upon reasonable request.

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
