# Peer review of "DNA Barcoding of Chironomid Larvae (Diptera: Chironomidae) from Large Rivers in South Korea to Facilitate Freshwater Biomonitoring and Public Health Surveillance"

_ijerph, 2022, doi:10.3390/ijerph191912035_

Round 1

Reviewer 1 Report

Dear Authors,

the presented work is interesting and up-to-date, especially for South Korea and could be very useful.

However, it needs to be clarified does the research offer more then just the know-how on how to identify chironomid larvae found in the water supply system.

The pictorial key is nice and practical, however it does not provide room for findings of the species that authors maybe did not collect during their research.

It also needs to be more clearly explained where do the data on the adult specimens come from.

All the additional comments and suggestions are given in the manuscript file.

Reviewer 2 Report

The work was performed using modern molecular methods and species identification using DNA barcodes. The identification of the species by comparing the COI barcodes allowed the authors to determine most of the obtained sequences. The comparison of the COI of the sequence makes it possible to identify only the at species level, while the identification of the genus, subfamily, etc. is doubtful. Therefore, the authors cannot accurately attribute sample H20L to genus Stenochironomus so it is worth adding the word «presumably» or «possibly». It is necessary to expand Table 1 with numbers from the GenBank which close to obtained samples, as well as the values of K2P distances between them.

Reviewer 3 Report

Dear authors,

this work is interesting and relevant to national authorities, well done. I am including my comments in the attached document.

My main concerns are:

1. some statements related to public health need to be backed by references otherwise it would sound as if you are saying them.

2. concerned on sample size as if these insects are so problematic, how come only 61 were found? While reading I was expecting to see a larger number, but at the end it is all linked to the sampling method.

3. the adults were mentioned and you set some as reference specimens but the information in this work is scarce. 

4. why was only one reference sequence used per species (including data mined from genbank; while BOLD was not considered)? and were these reference sequences published (i mean genbank sequence associated to a published manuscript).

wish you all the best with this work

regards

Round 2

Reviewer 1 Report

Dear Authors,

thank you for your valuable contribution to the studies of Chironomidae, I believe this manuscript will be useful for future research.

I added just a few suggestions in the revised version of the manuscript.

Author Response

Thanks for your useful corrections and helpful suggestions. We have corrected all the minor corrections and unclear sentences as in the attached final manuscript.

Unclear or confusing words and sentenses were corrected: For example, larval spill in drinking water ==> larval infestation in drinking water.

Table 1: The chironomid taxa were arranged by suggested classification system.

The manuscript was intensively edited and corrected by a native speaker.
